# First Findings of Buried Late-Glacial Paleosols within the Dune Fields of the Tomsk Priobye Region (SE Western Siberia, Russia)

**Alexandr Konstantinov [1]**, **Sergey Loiko [2,*]**, **Alina Kurasova [2]**, **Elizaveta Konstantinova [3]**, **Andrey Novoselov [3]**, **Georgy Istigechev [2]** and **Sergey Kulizhskiy [2]**

[1] Oil and Gas Geology Research and Educational Center, Tyumen Industrial University, Volodarskogo St. 56, 625000 Tyumen, Russia; konstantinov.alexandr72@gmail.com

[2] BIO-GEO-CLIM Laboratory, National Research Tomsk State University, Lenina St. 36, 634050 Tomsk, Russia; kurasovalina@gmail.com (A.K.); istigechev.g@yandex.ru (G.I.); kulizhskiy@yandex.ru (S.K.)

[3] Institute of Earth Sciences, University of Tyumen, Osipenko St. 2, 625002 Tyumen, Russia; konstantliza@gmail.com (E.K.); mr.andreygeo@mail.ru (A.N.)

* Correspondence: s.loyko@yandex.ru; Tel.: +7-903-913-0555

**Abstract:** Buried soils within aeolian deposits are considered an important tool for diagnosing, determining the age, and estimating the intensity of aeolian processes at the transition from the Pleistocene to the Holocene in the Northern Hemisphere. Late Pleistocene aeolian coversands and ancient inland dunes are widely distributed in the periglacial zone of Western Siberia. In contrast to the territories of Central and Eastern Europe, the paleosol archive of the aeolian sands and dunes of Western Siberia has not yet been studied. This paper presents the first findings of late Pleistocene paleosols within the ancient inland dunes in the southeast of Western Siberia (Ob–Tomsk interfluve, Tomsk region). The soils and their stratigraphic position were studied in the outcrop of the quarry, located in the junction zone of the second Tom river terraces and the ancient valley. Two types of paleosols were identified. The first one is confined to the central part of a small dune and is represented by a slightly developed Albic Arenosol with fragmentary humus horizon Ahb and a well-pronounced Eb. It can probably be considered as an analogue of the European Usselo soil. The second paleosol was found at the bottom of the interdune depression. It is represented by a brown Bwb horizon and probably corresponds to a Brunic Arenosol (Dystric). The second paleosol is characterized by a higher content of clay fraction and organic carbon, the presence of weak signs of illuviation, and richer and more diverse mineral composition. This soil is apparently an analogue of the European Finow soil. Radiocarbon dating of the charcoals found in the paleosols suggests that the first dates from the Younger Dryas (ca. 12,036 cal. yr. BP), and the second one from the Allerød (ca. 13,355 cal. yr. BP). The study results propose that the natural environment in the periglacial zone of the south of Western Siberia was generally similar to those in Central and Eastern Europe, and the activation of aeolian processes, which led to the formation of a dune relief, occurred at about the same time.

**Keywords:** dune fields; periglacial environment; Albic Arenosol; Brunic Arenosol; AMS-dating; Western Siberia

## 1. Introduction

Ancient inland dunes and aeolian coversands are widespread within the periglacial zone of Northwestern, Central, and Eastern Europe, as well as the Eastern European Plain [1–5]. The formation of aeolian sediments and relief occurred in areas with a wide distribution of sandy surfaces of

fluvioglacial, glaciolacustrine, and alluvial genesis. The primary formation of the dune relief took place in the Late Glacial and Preboreal [4,6]. During the formation of the aeolian relief in Europe, a mosaic of woody and herbaceous vegetation was maintained. This persisted through the Younger Dryas until forest biomes became dominant at 10 kyr cal. BP. In contrast, Siberian region featured predominantly treeless vegetation until 14–13 kyr cal. BP. A major change is seen around 14 kyr cal. BP. While graminoid-forb tundra is still common, in the Siberian region it tends to be replaced by low-high shrub tundra. There is a continuing increase of cold evergreen needle-leaved forest, and temperate deciduous forest appears at some sites in southern and central Europe [7]. The results of recent studies have shown that variations in climatic conditions throughout the Holocene, as well as anthropogenic activity, led to the stadial character of soil and landscape development in areas with dune fields and coversands [8–18].

The buried soils of the aeolian coversands and inland dune fields of the Northern Hemisphere are considered important stratigraphic boundaries that played a significant role for the timing of aeolian activity through the Late Pleistocene and Holocene. Two paleopedological marker horizons are distinguished for the late glacial fluvio-aeolian successions of Central and Eastern Europe. The first one is the Usselo soil (or Usselo horizon), described over a wide area of the Netherlands, northern Belgium, Denmark, northwestern Germany, and central Poland [19]. The horizons of the Usselo soil are characterized by low organic-matter content, the presence of charcoals, and the predominance of bleached quartz grains. Usselo soil, as a rule, has a thickness of 5–20 cm and corresponds to the Ahb and Eb horizons of an initial podzol or Albic Arenosol that was formed in good drainage conditions. The average age of Usselo-type soils varies from the Allerød to the Younger Dryas [20]. The second type, Finow soil, was first discovered near the city of Eberswalde-Finow 50 km northeast from Berlin [21,22]. The soil consists of a 5–30 cm thick brownish horizon that contains charcoals. The widespread distribution of this type of paleosol was later established for the regions of western [23] and northern Poland [19]. The origin of the Finow soil is still debatable. According to some points of view, this layer is the sediment associated with the impact hypothesis [24], but not the Bwb horizon corresponding to a Brunic Arenosol. Based on the results of OSL and radiocarbon dating, development of Finow soils occurred during approximately the same period as the Usselo soil. In contrast to Central Europe, the findings and detailed descriptions of paleosols within the fluvio-aeolian successions of the European part of Russia are rather rare [6,12].

The territories located to the east of the Ural Mountains are also characterized by the wide distribution of ancient inland dunes and aeolian coversands [25–31]. The formation of aeolian sandy deposits and adjacent dune relief of Western Siberia occurred in the arid cold climate conditions of a periglacial zone [32,33] and, apparently, this process took place synchronously with other regions of Northern Eurasia. However, Late Pleistocene inland dunes and coversands of the south of Western Siberia are very poorly studied compared with their counterparts in central Europe and on the Eastern European Plain. The exact time of primary dune formation, as well as the intensity of aeolian activity in the Holocene, are unexplored, and paleosols have not been discovered. The following paper attempts to fill this gap.

This article presents the first results of studying buried soils of ancient inland dune fields in the southeastern part of Western Siberia, found in the outcrops of Tahtamyshevo quarry near the city of Tomsk. The main aims of our study were: (1) a description of the stratigraphic position of buried soils and the determination of their age; (2) analytical characteristic of their properties; (3) meso- and micro-morphological description; and (4) placing the buried soils of this study within the context their possible analogues from fluvio-aeolian successions of Central and Eastern Europe, and Eastern European plains.

## 2. Materials and Methods

### 2.1. Regional Settings

The study area is located in the southeastern part of the West Siberian Plain near the foothill borders of Altai–Sayan mountainous country. The territory with a wide distribution of inland dune fields is confined to the Ob–Tomsk interfluve. The Ob–Tomsk interfluve is a plain with absolute altitudes of 110–190 m, bordered by multilevel terraces of Ob and Tomsk Rivers in its marginal parts. A number of ancient river valleys (or possible spillways) intersect the interfluve from southwest to northeast [34].

The research objects are located in the eastern, peripheral part of the interfluve (Figure 1), where aeolian coversands and dune fields are common within the ancient valleys and on the surfaces of low fluvial terraces. Linear dunes stretch from the northeast to the southwest and, as a rule, have a relative height of about 10 m, much less than 20 m. The results of studies performed by S.V. Parnachev showed that the typical section of sandy sediments in the areas with pronounced dune relief consists of aeolian and fluvial units [30]. The aeolian genesis of the upper part of the sediments composing the dune ridges was established based on sedimentological features.

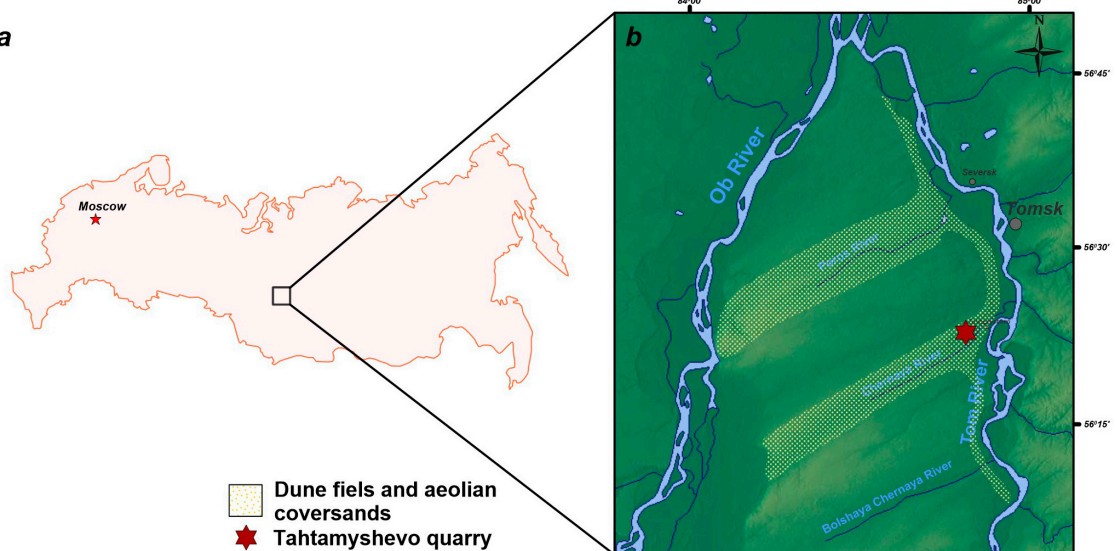

**Figure 1.** Location of the study area: (**a**) within the territory of Russia; (**b**) within the Ob–Tom interfluve.

The modern climate of the study area is continental. The average annual temperature is 0.6 °C and the average annual rainfall is near 550 mm [35]. The study area belongs to a zone of hemiboreal forests. Green moss pine forests and cowberry green moss forests represent the modern vegetation of the ancient river valleys and low terraces covered with aeolian sands. The bottoms of large interdune depressions are often swamped. Albic Arenosols are the most common soils within the dune ridges, while Brunic Arenosols and Entic Podzols occupy gentle slopes and bottom parts of interdune depressions [36]. The territory under consideration is characterized by a long history of anthropogenic activity. This is evident due to the numerous archaeological monuments confined to the sites with dune relief [37].

### 2.2. Site Description and Sampling

The buried soils were found in the outcrops of a sand quarry located in the outskirts of Tahtamyshevo village (56°22′22.01″ N, 84°50′14.76″ E). The quarry occupies the surface of the second Tom River fluvial terrace in its junction with the Chernorechenskaya ancient valley. The aeolian relief is better expressed in the northern part of the quarry while, in the southern part, the dunes disappear

and the flat bottom of the ancient valley begins. The absolute heights vary from 100 to 110 m in the northern part of the quarry, and from 87 to 100 m in the southern part.

Two sections with buried soils were selected for detailed studies (Figure 2). The first soil was found in the section on the eastern steep edge of the quarry, revealing a small dune 5–6 m high and 100 m wide; the occurrence interval varied, from 165 to 180 cm from the surface of the dune. The second section with soil was confined to a flat surface between two dunes and was found in the northwestern part of the quarry. To the north of there, the dunes are much higher than those located to the south.

Samples of sandy sediments were collected with an interval of 30–40 cm from two sections with buried soils. Since the buried soils showed rather high horizontal variability and individual horizons often consisting of separate patches with diffuse boundaries, samples of paleosols were collected from at least three points located at the same level (for each distinguished horizon). Undisturbed samples that were later used for the preparation of transparent thin sections (2.5 × 4.5 cm) were collected from the lower and upper parts of the second buried soil. Charcoal samples for AMS-dating were taken from the Ahb + Eb (first section) and Bwb horizons (second section).

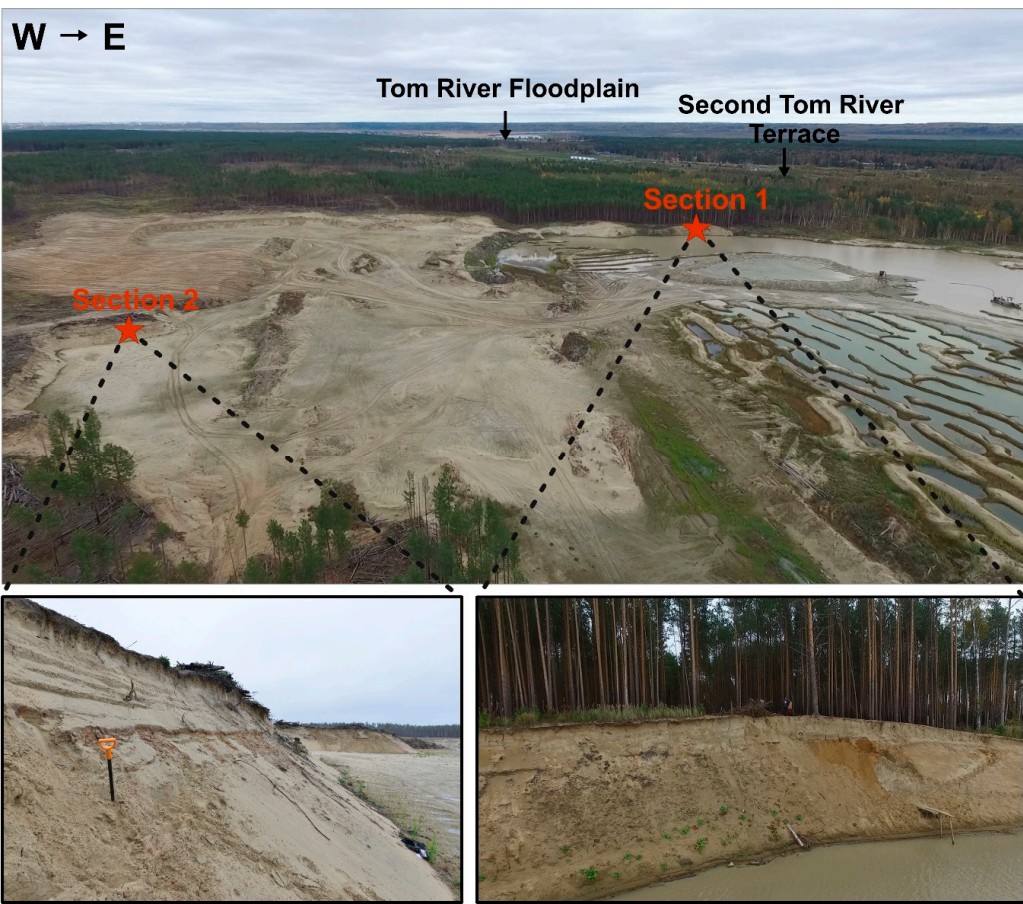

**Figure 2.** Location of sections with paleosols within the territory of Tachtamishevo quarry (photographs made by drone).

## 2.3. Analytical Methods

The texture of studied soils and sandy sediments was determined using an LS 13 320 Beckman Coulter (Beckman Coulter, Indianapolis, USA) analyzer with the preliminary dispersion of samples happening with sodium pyrophosphate and sonication. The color of the samples was measured in the Munsell system using a vs. 450 X-Rite (X-Rite, Michigan, USA) spectrophotometer. Preliminary mesomorphological studies were performed using a Stemi 2000 stereomicroscope (Carl Zeiss

Microscopy GmbH, Oberkochen, Germany). Micromorphological studies in transparent thin sections were carried out with an Eclipse LV100POL (Nikon, Tokyo, Japan) polarization microscope at magnifications from 0 to 40. The description of thin sections and individual elements of the microstructure was carried out according to [38]. Microscopic and submicroscopic studies were performed using a TM3000 (Hitachi, Tokyo, Japan) scanning electron microscope with an X-ray attachment for elemental analysis of the Quantax 70 surface at magnifications from 40 to 2000.

Chemical analyses were performed according to generally accepted methods [39]. The following properties were determined for the studied paleosol samples: pH $H_2O$ and pH KCl by a potentiometric method in suspension with a soil/liquid ratio of 1:2.5. Fed was extracted with sodium dithionite-citrate-bicarbonate and Feo with an ammonium oxalate buffer solution [39]. The concentrations in the extracts were determined spectrophotometrically using a SmartSpec Plus (Bio-Rad Laboratories, Inc., Hercules, California, USA) spectrophotometer. We also determined the loss on ignition, and organic carbon and total nitrogen content using a Flash 2000 Thermo Scientific organic elemental analyzer.

The radiocarbon ages of charcoals from both paleosols were obtained using accelerator-mass spectrometry (AMS) at the Radiocarbon Dating and Electron Microscopy Laboratory of the Institute of Geography of the Russian Academy of Sciences, and at the Center for Isotopic Studies at the University of Georgia. Age calibration was performed using the CALIB REV7.1.0 software package based on calibration curves [40].

## 3. Results

### 3.1. Stratigraphic Position of Paleosols

The photographs and the scheme showing the stratigraphic position of the paleosols from the two studied sections, as well as their textural variability are given in Figures 3 and 4.

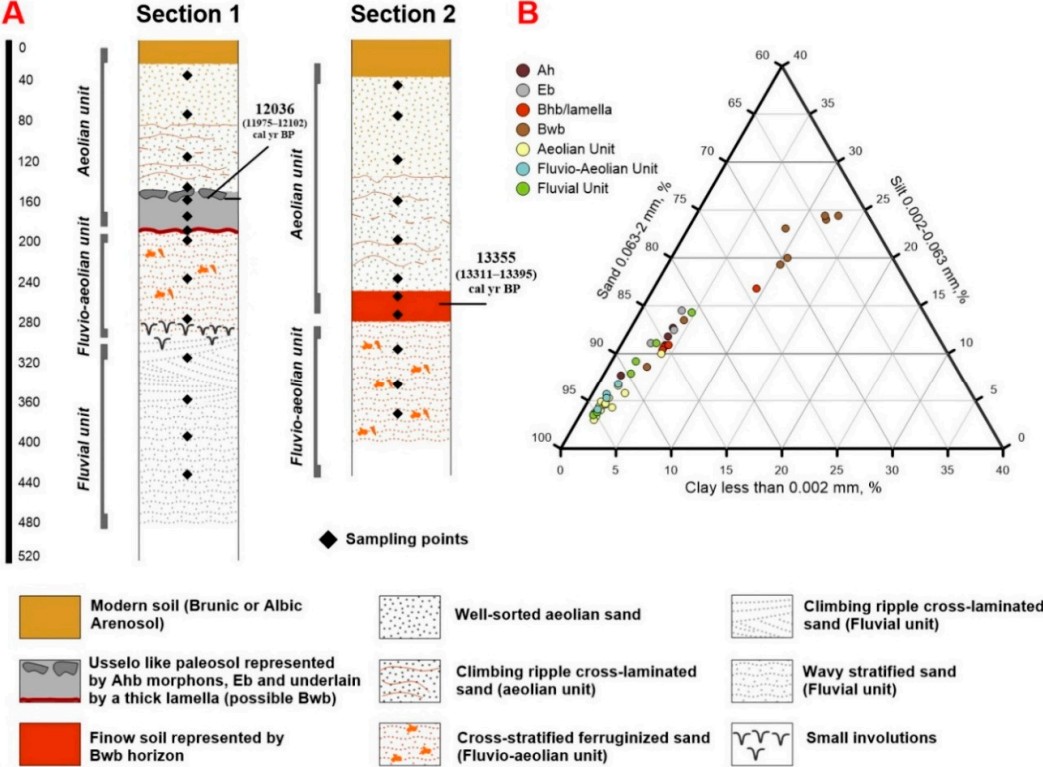

**Figure 3.** Stratigraphic and sedimentological characteristics of a fluvio-aeolian succession: (**A**) scheme showing stratigraphic position of paleosols in studied sections; (**B**) ternary diagram illustrating textural variability of studied sediments and paleosols.

The first section reveals a small linear dune with a relative height of 5–6 meters, oriented from the northeast to the southwest. The dune is located in the periphery zone of the aeolian relief: in the south of this site, dunes transform into a flat sheet of aeolian sands, covering the fluvial deposits of the second terrace. Three contrasting units were distinguished in the composition of the sediments of the first section: aeolian, lying directly under the thin modern soil (40–220 cm), fluvio-aeolian (220–290 cm), and fluvial (290–440 cm). Such stratigraphy is generally characteristic for Late Pleistocene fluvio-aeolian successions of Eastern Europe [12,41]. Strata of well-sorted light-yellow fine-grained sands represent the aeolian unit. Numerous thin winding lamellae are distinguished in the depth interval of 80–150 cm. Weakly expressed aeolian lamination that is strongly disturbed by roots appears at the depth of 95 cm. Paleosol 1, underlain by a thick lamella, occurs in the interval between 155 and 180 cm. Greyish-ocherous laminated fine-grained sand, practically not different from sediments of the middle part of the aeolian unit in terms of grain size, forms a transition to the lower unit. Fine-grained, iron-coated rusty fluvio-aeolian sand with weakly inclined stratification and small involutions, possibly indicating the degradation of permafrost [42], lie in the interval from 220 to 300 cm. Gray-bluish fine-grained sands with well-pronounced horizontal stratification and rare iron-smudging spots represent the underlying fluvial unit. The sediments of the fluvial unit differ from the overlying sands by a slight increase in the content of the fraction of 0.05–0.1 mm. Paleosol gradually disappears in the marginal parts of the dune while underlying lamellae become thinner and stretch, disagreeing with the inclination of the slope.

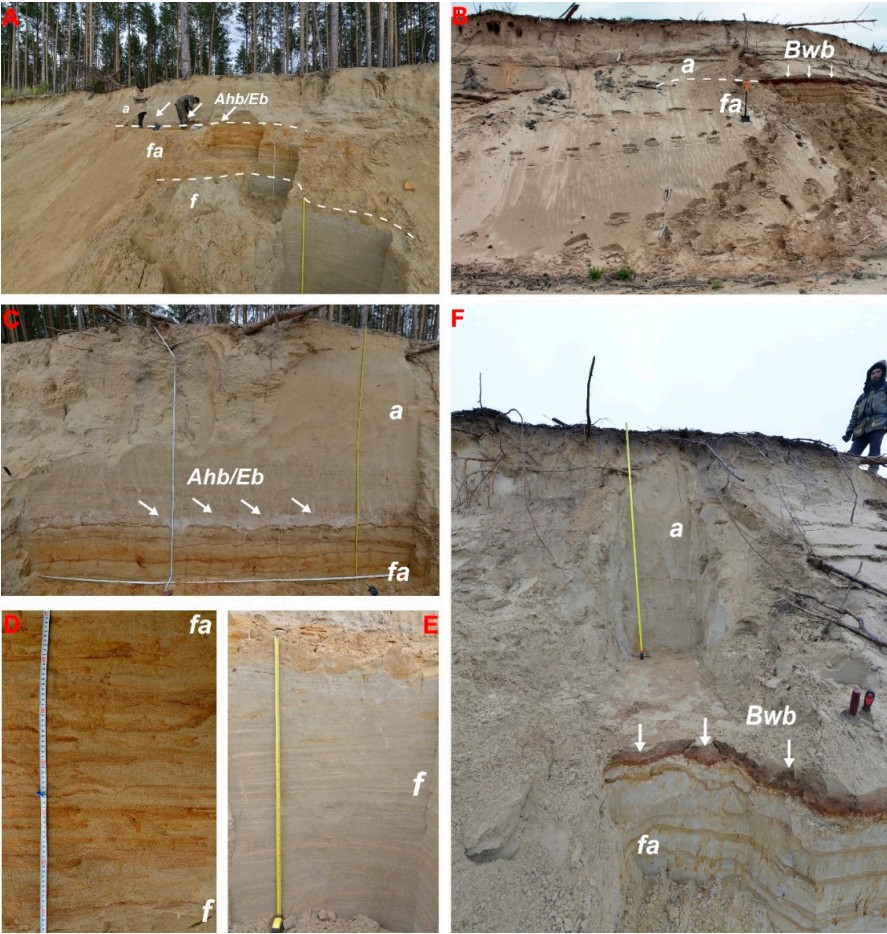

**Figure 4.** Fluvio-aeolian successions and sedimentary features (*a*—aeolian unit, *fa*—fluvio-aeolian unit, *f*—fluvial unit): (**A**) section 1; (**B**) section 2; (**C**) aeolian unit with Usselo-like paleosol, section 1; (**D**) fluvio-aeolian unit, section 1; (**E**) fluvial unit, section 1; (**F**) Finow-like paleosol on the contact of aeolian and fluvio-aeolian units in section 2.

The second section is characterized by a similar structure. An aeolian unit occurs at depths between 50 and 260 cm, directly under the modern soil (Brunic Arenosol). Strata of fine-grained pale-yellow sands without pronounced layering represent the upper part of the aeolian unit. In the interval of 130 to 260 cm, pale-gray sands replace yellow sands. The fuzzy aeolian stratification becomes better expressed in the lower part of the unit. Winding thin lamellae with numerous signs of disturbances, mainly due to root penetration, occur within this stratum. Paleosol 2 is confined to the border of the aeolian unit and underlying fluvio-aeolian sediments, and lies in the 260–278 cm interval. It consists of two rather contrasting subhorizons, Bwb, g (260–268 cm) and Bwb (268–278 cm). The difference between them is in the presence of gray patches in the upper horizon. Bluish-gray fine-grained sands with separate patches enriched with mafic material, and rare involutions alternating with reddish-brown weakly cemented 1–2 cm thick interlayers, represent sediments of the fluvio-aeolian unit, underlying the paleosol. The fluvial unit was not observed in this section. The modern soil at the top of the section is located in a subordinate position, at the foot of the dunes. The buried Paleosol 2 soil lies at the maximum depths at the study site. On the adjacent elevated areas, Paleosoil 2 rises to the modern soil surface, up to a depth of 150 cm or even less, and gradually breaks into a series of lamellae or fully disappears near dune ridges.

*3.2. Morphology of the Studied Paleosols*

Paleosoil 1 (Section 1) is similar to Albic Arenosols (Figure 5a,c), which are now abundant in areas with the dune relief within the Ob–Tomsk interfluve. It is possible to distinguish its likeness with the Usselo soil that marks the boundary of the aeolian unit in the fluvio-aeolian successions of Central and Eastern Europe. The Ahb horizon, represented as separate patches, most clearly expressed in the central part of the outcrop lies at a depth of 155–165 cm from the surface. Unlike European analogues, the spots of the humus horizon are much lighter and contain less humus (10YR 7/2). The border with the underlying eluvial horizon, expressed in the fragmentation of Ahb, is rather conventional. Therefore, it is possible to count Eb and Ahb as a single horizon (Ahb + Eb). Charcoals for AMS dating were collected at the border of Ahb and Eb horizons in the left side of the outcrop. Charcoals were small (up to 1.5 mm) and rounded. The Eb horizon is much better expressed and looks like a bleached spotted eluvial layer (10YR 7/1), with a capacity of 8–12 cm. The Eb material is represented by washed well-sorted quartz grains that show no difference from the enclosing aeolian sediments in terms of granulometry. Gray and brownish-red spots, which may correspond to the material from underlying (Bwb) or overlying (Ahb) horizons, as well as very thin illuvial lamellae represented by separate 1–3 cm long segments, grouped into winding chains of 10–15 cm length, are noted within the horizon. The Eb lies on a dense, 2–3 cm thick, lamella. Unlike the underlying lamellae, it has a greater thickness, a richer color (10 YR 6/3), and it is noticeably denser. Therefore, it is possible to suggest that these lamellae developed over the Bw horizon.

Paleosoil 2 (Section 2) is represented by a single Bwb horizon with a thickness of 12–18 cm. Its upper part may be subdivided as individual subhorizon Bwb, g (Figure 5b,d). The material of the buried soil has a brownish color (10 YR 5/4). The upper part of the horizon has a pinkish shade; the lower part is slightly reddish. Mineral grains are covered with clayey and organo-ferruginous coatings. Paleosoil 2 has a higher density granulometric composition than host deposits, as well as very weak signs of a blocky subangular structure. The border with the overlying aeolian stratum is abrupt, smooth, or slightly wavy. The border with the underlying fluvio-aeolian unit is less pronounced and gradual for at least half of the uncovered interval. Dark-colored inclusions and very fragile small charcoals sized 1–2 mm, were found in the lower part of the horizon. Large disintegrated and angular charcoals up to 1 cm that were selected for further AMS dating occurred much more rarely (Figure 5d,e). The presence of a well-defined Bwb horizon in Paleosol 2 allows us to consider it as an analogue of the Finow Soil.

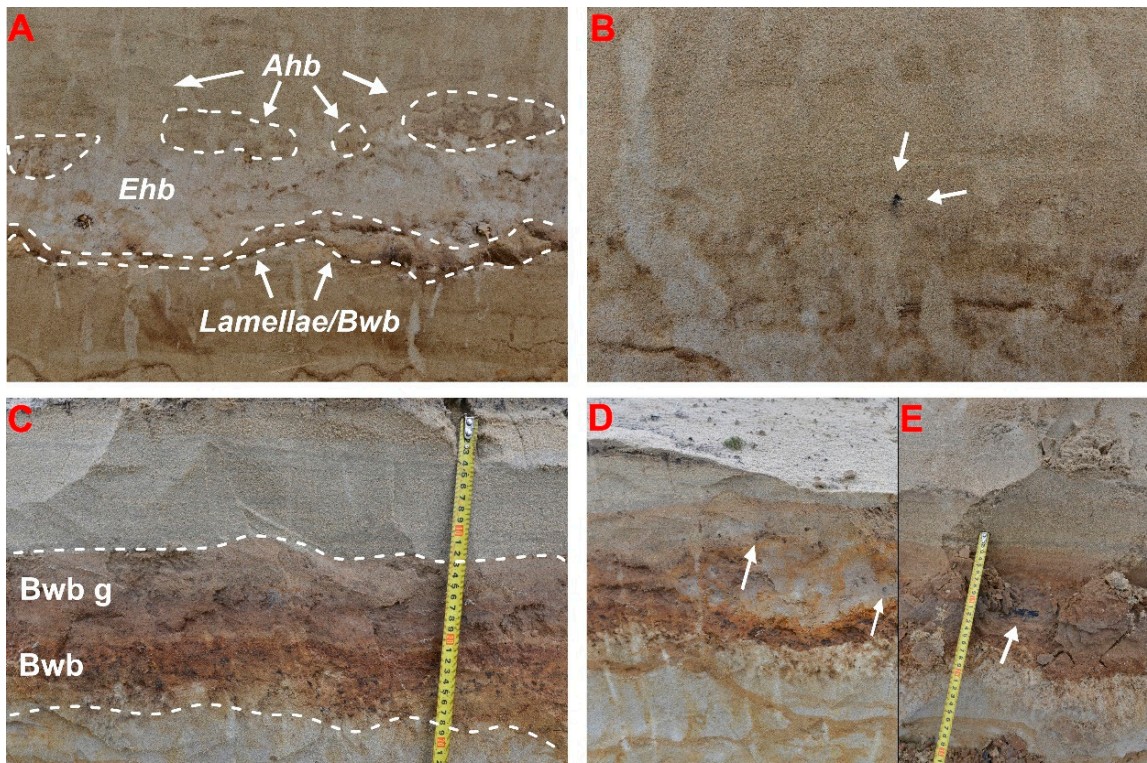

**Figure 5.** Morphological features of studied paleosols: (**A**) separated morhons of Ahb, Eb, and thick lamellae possibly developed over Bwb horizon of Ussel-like paleosol in section 1; (**B**) stratified Bwb horizon of Finow-like paleosol in section 2; (**C**) large charcoal in Ahb horizon, section 1; (**D,E**) charcoals in Bwb horizon, section 2.

### 3.3. Properties of Paleosols

The data on the particle-size distribution and chemical properties of paleosols are given in Tables 1 and 2. The Ahb and Bwb/lamellae horizons of Paleosol 1 are slightly enriched in clay and silt fractions, while in the Eb the content of these fractions is comparable or slightly lower (for silt) than in the enclosing aeolian sands. Paleosol 2 has a higher density granulometric composition: the content of clay in a central part of the Bwb horizon is 12%, the silt fraction is 18%, and their sum is 30%. Samples from the Bwb horizon can be defined as loamy sand, which is significantly different from all other studied samples that were identified as sand, in accordance with the USDA (Figure 3b).

**Table 1.** Texture of buried soils.

| Horizons | Depth, cm | Grain Size Fractions, % | | | | | | | | |
|---|---|---|---|---|---|---|---|---|---|---|
| | | >0.0002 mm | 0.0002–0.002 mm | 0.002–0.02 mm | 0.02–0.05 mm | 0.05–0.1 mm | 0.1–0.25 mm | 0.25–0.5 mm | 0.5–1.0 mm | 1–2 mm |
| | | | | | Paleosol 1 | | | | | |
| Ceol | 140–155 | 0.3 | 1.3 | 2.3 | 0.2 | 5.5 | 79.5 | 6.0 | 5.0 | 0.0 |
| Ahb | 155–165 | 0.6 | 3.2 | 6.6 | 0.8 | 14.6 | 69.1 | 3.0 | 2.0 | 0.0 |
| Eb | 165–176 | 0.3 | 0.9 | 1.4 | 0.2 | 6.9 | 81.1 | 5.0 | 4.2 | 0.0 |
| Bwb/lamellae | 176–180 | 1.2 | 8.1 | 12.9 | 0.5 | 8.8 | 63.7 | 3.3 | 1.5 | 0.0 |
| C | 180–190 | 0.5 | 1.3 | 1.9 | 0.4 | 8.8 | 80.8 | 3.8 | 2.6 | 0.0 |
| | | | | | Paleosol 2 | | | | | |
| Ceol | 240–260 | 0.3 | 1.4 | 2.4 | 0.2 | 6.4 | 76.9 | 7.3 | 5.1 | 0.0 |
| Bwb | 260–267 | 0.6 | 3.8 | 7.9 | 0.8 | 13.3 | 66.8 | 3.4 | 2.3 | 1.1 |
| Bwb | 267–271 | 1.3 | 10.7 | 16.8 | 1.3 | 12.9 | 53.6 | 2.1 | 1.3 | 0.0 |
| Bwb | 271–276 | 1.3 | 9.2 | 12.2 | 1.4 | 13.2 | 58.3 | 2.8 | 1.6 | 0.0 |
| C | 276–296 | 0.3 | 1.4 | 2.4 | 0.2 | 6.4 | 76.9 | 7.3 | 5.1 | 0.0 |

All studied samples of buried soils are acidic or weakly acidic (Table 2). Values of pH $H_2O$ slightly vary from 5.8 to 6.0 in the horizons of Paleosoil 1, while pH KCl varies over a wider range: from 5 to 5.7 in Ahb, from 4.1 to 4.5 in Eb, and 4.4–4.5 in Bwb/lamellae. The Bwb horizon of Paleoscale 2 has large pH variability: the upper and middle pH $H_2O$ is 5.9–6.0, decreasing in the lower part to 5.4–5.5. The KCl pH decreases from 4.9–5.0 in the upper part to 4.1–4.4 in the middle and lower parts.

Loss on ignition is 2.0%–2.4% in the Ahb, 1.5%–1.9% in the Eb, and 1.9%–2.7% in the Bwb/lamellae horizons for Paleosoil 1 (Table 2). Higher values are characteristic for the Bwb horizons of Paleosoil 2: from 2.5%–2.7% in the top and middle, to 2.9%–3.7% at the bottom. The content of organic carbon in the first soil was insignificant: 0.4–0.8 g $kg^{-1}$, and the C/N ratio varied significantly, from 1.2 to 6.5. In Paleosol 2, these values were higher: 0.9–1.6 g $kg^{-1}$ for C, and a 3.5–6.1 C/N ratio. Higher contents of both Feo and Fed were observed in Paleosol 2, which can be explained because of higher content of clay fraction.

**Table 2.** Chemical properties of buried soils.

| Horizons | Depth, cm | Color | LOI, % | C org, g $kg^{-1}$ | N, g $kg^{-1}$ | C/N | pH $H_2O$ | pH KCl | Feo, g $kg^{-1}$ | Fed, g $kg^{-1}$ |
|---|---|---|---|---|---|---|---|---|---|---|
| Paleosoil 1 | | | | | | | | | | |
| Ahb | 155–165 | 10YR 7/2 | 2.0–2.4 | 0.4–0.7 | 0.3–0.5 | 1.2–1.4 | 5.8–6.0 | 5.0–5.7 | 0.01–0.10 | 1.3 |
| Eb | 165–167 | 10YR 7/1 | 1.5–1.9 | 0.6–0.8 | 0.1–0.2 | 3.0–6.5 | 5.7–6.0 | 4.7–5.0 | 0.01–0.02 | 0.5 |
| Bsb/lamellae | 167–177 | 10YR 6/3 | 1.9–2.7 | 0.6–0.8 | 0.3 | 1.8–2.6 | 5.9–6.0 | 5.7–4.9 | 0.15–0.34 | 5.0 |
| Paleosoil 2 | | | | | | | | | | |
| Bwb | 240–247 | 10YR 5/4 | 2.6–2.7 | 1.4 | 0.3 | 3.5 | 6.0–6.1 | 4.9–5.0 | 1.09–1.20 | 16.5 |
| | 247–251 | 10YR 5/4 | 2.5–2.7 | 1.3–1.4 | 0.3 | 4.6–4.7 | 5.9–6.0 | 4.1–4.5 | 0.90–0.94 | 10.8 |
| | 251–256 | 10YR 5/4 | 2.9–3.7 | 0.9–1.6 | 0.2–0.3 | 5.7–6.1 | 5.4–5.5 | 4.4–5.5 | 0.88–1.01 | 11.2 |

### 3.4. Micromorphology of Paleosols

Well-sorted, fine-grained quartz-feldspar sands with single rock fragments, amphibole grains, and hematite and mica particles represent the material of the Ahb and Eb horizons of Paleosoil 1. Quartz and coarse feldspar grains are often characterized by close-to-perfect roundness, which is generally characteristic of the aeolian sands of the Ob–Tomsk interfluve [30]. Organo-ferruginous coatings and adhered particles of fine silt are noticeable on the surfaces of individual grains in the Ahb horizon. Coatings on grains in the Eb horizon are almost absent. Angular mineral grains from the fraction of coarse silt and very fine sand often have traces of crushing (Figure 6).

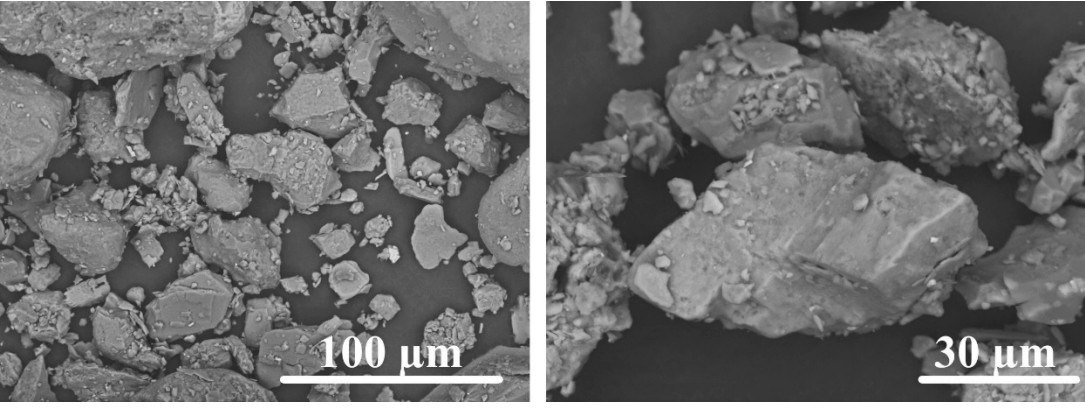

**Figure 6.** Angular silty particles from Ahb horizon of paleosol 1 with signs of cryogenic crushing.

The upper (Bwb, g) and lower (Bwb) subhorizons of Paleosoil 2 have significant differences in microstructure (Figure 7). Coarse material prevails over the matrix in the upper (Bwb, g) subhorizon of Paleosol 2 (Figure 8). The matrix is unevenly distributed and consists of clay material. The upper part

of Bwb is characterized by enaulic and locally gefuric, close to chitogefuric c/f-related, distribution, which is associated with the characteristic features of sedimentation in periglacial conditions [43] and is typical in almost all Bwb horizons of the Finow soils of Central and Eastern Europe [19]. Well-oriented thin clay coatings and bridges are noticeable around mineral grains. Signs of several phases of illuviation, possibly associated with the movement of fine fractions after soil burial, were found in some places. Ferruginous hypocoatings over clay coatings and bridges indicate periodic conditions of waterlogging and changes of redox conditions, which is proven by the subordinate position of this section in both modern and periglacial relief. Quartz grains (40%–55%), feldspars particles (up to 15%), and rock fragments (30%–35%) are the most common components of coarse material. Sorting of the material is worse than in overlying aeolian sediments, and varies from medium to weak. Most of the grains are well-rounded, though oblong fragments are also found. Many grains on the surface have traces of dissolution. Feldspars are represented by microcline, plagioclase and orthoclase. Some grains are pelletized; some have traces of leaching, along which ferruginous clay material develops. Well-rounded fragments of metamorphic rocks mainly quartzite, represent fragments of rocks. Slightly rounded elongated fragments of mica-quartz mudstones occur less commonly. Single grains of myrmekite were detected, which indicates the presence of effusive rocks. Single grains of epidotes represent accessory minerals.

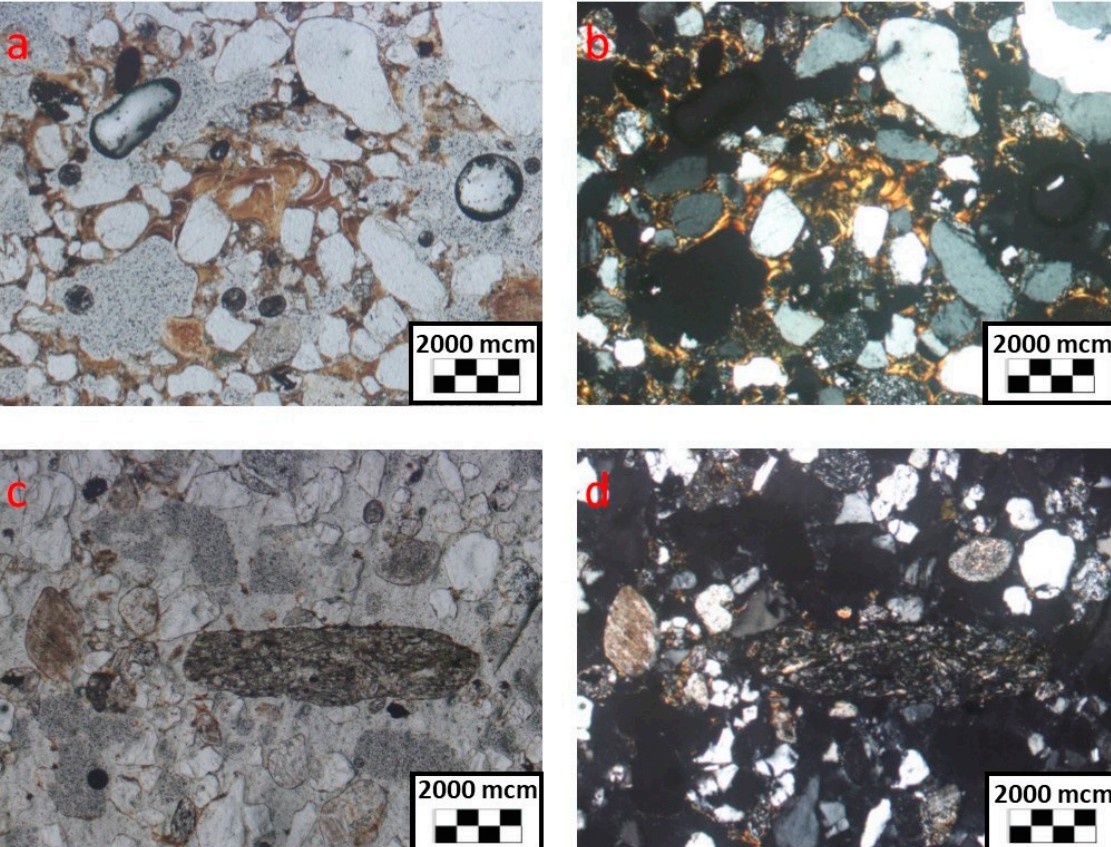

**Figure 7.** Micromorphological features of the upper and bottom parts of Bwb horizon (Paleosol 2): (**a**,**b**) Light brown, speckled fine mass, equal to single spaced fine enaulic (partly chitonic) related distribution in upper part of Bwb (PPL (plane polarized light) and XPL (crossed polars), respectively); (**c**,**d**) almost absence of clay and coarse fragments of rocks in the bottom part of Bwb horizon (PPL and XPL, respectively).

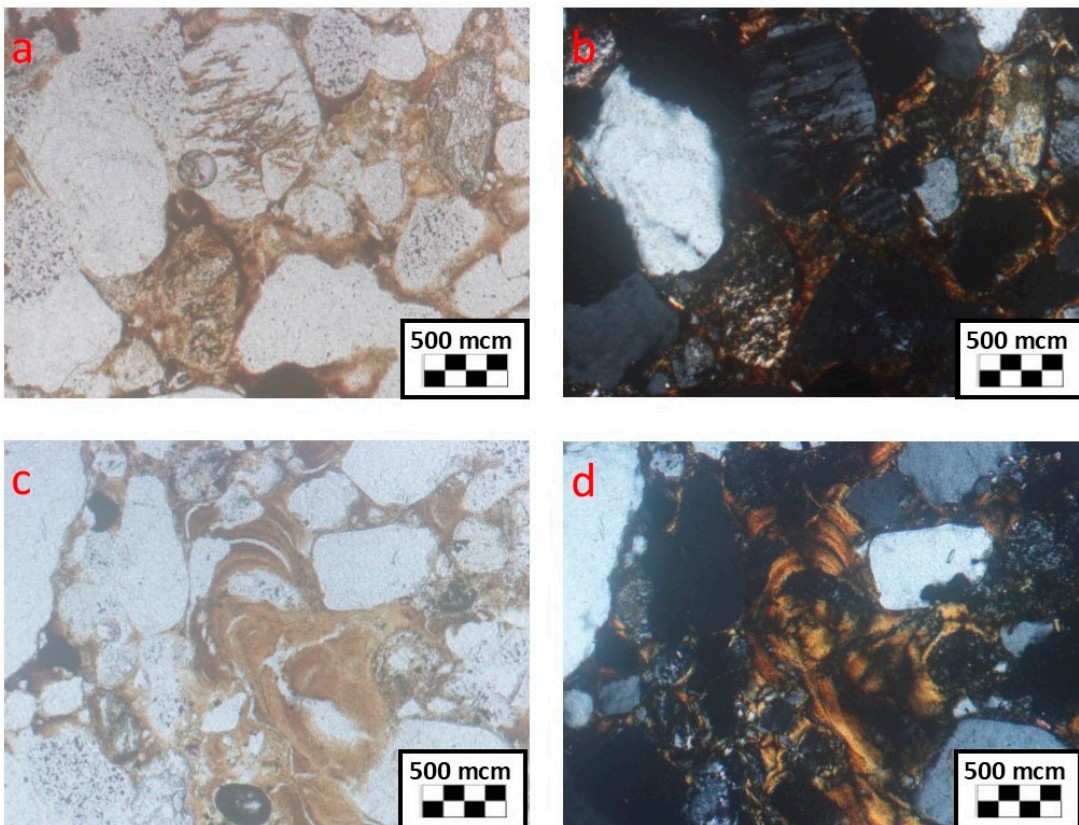

**Figure 8.** Micromorphological features of the upper part of Bwb horizon (Paleosol 2): (**a,b**) clay coating and bridges (PPL and XPL, respectively); (**c,d**) two phases of clay illuviation (centre): limpid yellow-brown clay coating (older) and dusty brown clay coating with ferruginous hypocoatings (possibly younger) (PPL and XPL, respectively).

The lower (Bwb) subhorizon differs from the upper one. The matrix is practically absent except for thin clay coatings on the surface of individual grains. Coarse fragments are represented by quartz (more than 50%) and feldspar (25%–30%) grains. However, there are large fragments of clay–quartz mudstones and single quartzite grains. The sorting of the material varies, from weak to medium. In general, grains are well-rounded, although both isometric and elongated grains are found at the same time. Feldspars are mainly plagioclase and orthoclase, and, less commonly, microcline. Accessory minerals are represented by single grains of tourmaline and, often, highly modified epidote.

*3.5. Dating*

The results of determining the numerical age of charcoals (Table 3) showed that the formation of Paleosoil 1 at least partly occurred in the middle of the Younger Dryas, and Paleosol 2 probably was formed earlier, in the middle of the Allerød. Still, this data is rather controversial, as well as the exact period of soil formation determined only by OSL dating and radiocarbon ages obtained by dating charcoals can be considered as a hint of soil age and more likely to the time reflecting the end of the soil's exposure to the surface. The obtained data are in good agreement with the results of determining the age of Usselo and Finow horizons for the territories of Central and Eastern Europe, most of which also relate to the Allerød (12,650–13,900 cal. yr. BP) or the Younger Dryas (11,550–12,650 cal. yr. BP) and much less frequently to the Early Holocene [19]. The age of Paleosoil 1 reflects the time of the activation of aeolian processes, and their burial. The age of Paleosoil 2 indicates that soil formation within the territory under consideration began in the Allerød, and, possibly, even earlier, in the Ancient Dryas [44].

**Table 3.** The results of dating charcoals of Paleosols.

| Sample | Lab Number | Sample Type | $^{14}$C, BP (1σ) | pMC, % | Age cal yr BP (1σ) |
|---|---|---|---|---|---|
| Section 1 Ahb/Eb | IGAN$_{AMS}$-6085 | Charcoal | 10270 ± 30 | 27.857 ± 0.097 | 12036 (11975–12102) |
| Section 2 Bwb | IGAN$_{AMS}$-6084 | Charcoal | 11510 ± 30 | 23.860 ± 0.090 | 13355 (13311–13395) |

## 4. Discussion

### 4.1. Paleosols and the Environment of Their Formation

In both sections, fluvial (fluvio-aeolian) sediments underlie paleosols. At present, these surfaces lie 15–20 m higher than the low-water levels of Tom and Ob, forming fluvial terraces. There are no obvious interruptions between parent material paleosols and fluvial sediments. Because the soils date to the Allerød, the youngest terraces are older than Allerød. Normally, alluvial deposits of low terraces are covered with loams that have a thickness of 0.5 to 10 (15) m. The deposition of these loams was completed by the Younger Dryas, about 12,560 cal. yr. ago. Variants with the absence of loamy sediments on the surface of low terraces, as in the sections under consideration, are associated with high aeolian activity that prevented the accumulation of aeolian sand prevented the accumulation of finer grained sediments. However, during periods of stabilization, as well as under hydromorphic conditions, subaerial material was deposited as layers with heavier granulometry within the strata of fluvial and fluvio-aeolian sandy sediments. The Bwb horizon of Section 2 probably includes such parent material.

The gradual attenuation of river activity at the end of the Pleistocene, established by the reduction of the size of the channels during the transition from the Pleistocene to the Holocene [45], led to the formation of chains of separated water bodies, periodically communicating during floods. Synchronous aeolian activity resulted in the formation of fluvio-aeolian sediments that underlie paleosols. At a certain moment, these residual reservoirs became temporarily dry in the summer season, and the accumulation of bottom sediments began. This hypothesis is based on the fact that the signs of gleying in Paleosoil 2 are not ubiquitous. Such hydromorphic soils were probably the basis for the further development of Paleosoil 2 that marks the bottom of depression. The material of Paleosol 2 has a higher density granulometric composition that leads to the conclusion that it has a subaerial origin and was formed synchronously with the upper part of loess loams that are widely distributed within the low above-floodplain terraces. The presence of effusive rocks in Paleosoil 2 indicates a still-persistent connection of local depressions with the riverbed during the flood period. We mean that these soils developed partially in times of sediment inflow to the surface. The presence of large charcoals indicates that elevated-landscape positions were covered with forest vegetation that was subjected to frequent fires.

Along the edges of the ancient depression, where Paleosoil 2 approaches the surface, it stratifies into several lamellae that rise practically to the modern surface. This probably indicates a periodic colluvial transport of sand downslope, which is accumulating in the toe slope position. Round gley spots with an ocher rim in the upper part of the Bwb, g subhorizon probably indicate wood falling to the bottom of the depression. Therefore, it is possible to suggest that Paleosoil 2 operated in water saturation conditions, during which time clay coating was formed (Figure 7). The relatively warm climate of the Allerød could have also contributed to textural differentiation [46,47]. We do not exclude the possibility of the formation of clay illuviation in the subsequent periods of the Allerød or even Holocene, after the burial. Since the relief of the surface where paleosol was found is concave, temporary perched water can move along it, laterally transporting clay particles. The possibility of such a process was experimentally shown for the territory under consideration [48].

The burial of Paleosoil 1 is based on the results of AMS-dating the charcoals that occurred at about 12,036 cal. years. Forest fires were probably one of the reasons for the activation of aeolian processes. The soil was formed in the conditions of a warmer Allerød. However, the presence of quartz grains with clearly visible chips and traces of crushing in the fractions of coarse silt and very fine sand of Ahb

can be an indirect indicator of quite intense frost weathering in cold climate conditions [49]. Therefore, Paleosol 1, at the last stage of its development, probably persisted in the Younger Dryas for quite a long time, which is consistent with the dating. Paleosoil 1 was formed in the conditions of a flatter topography than the modern one. The later relief was complicated by dunes.

The formation period of the considered paleosols could coincide in time; perhaps they could be different parts of a single paleocatena. The presence of an Eb horizon in the first paleosol and signs of illuviation in the second, lying hypsometrically below, support this hypothesis. Both paleosols could remain until the Younger Dryas—until the aeolian activity in the middle of this period did not transform the relief. This could be due to the increased severity of the climate that occurred in the Younger Dryas. Some research works [46] indicate the possibility of permafrost presence during this period. However, we did not observe any examples of permafrost structures in the studied paleosols.

The data on the age and morphology of studied soils are in good agreement with the results of paleoenvironmental reconstructions of the natural environment of Western Siberia in the Late Pleistocene. Other studies also indicate a cold climate and the presence of dry periods within the territory under consideration, when aeolian processes were activated [32,50].

When did the final stabilization of the dune relief on the Ob–Tomsk interfluve take place? Aeolian activity was still recorded after 10.5 thousand cal. yr., judging by the presence of sandy sediments without pollen in Lake Kirek, younger than a specified age [51]. It is believed that with the onset of the Holocene there was climate warming, which gradually reached its optimum by the Atlantic period. The territory of the Ob–Tomsk interfluve acquired a forest appearance in the range of 9–8 thousand cal. yr. in [46]. Sapropel deposits in lakes began to form after 8–7.5 thousand cal. yr. [51]. The solid flow of Ob and Tom Rivers stabilized near 9.5 cal. yr., and sandy alluvium was replaced by silty sediments, and peat accumulation began in the oxbows [52]. These dates coincide with data on the ancient dunes of the Upper Volga Basin [12], which have a range of 9–8 thousand years, corresponding to 10.6–8.8 cal. yr. There are opinions that this stabilization of dunes is associated not only with climate change, but also with the restructuring of ecosystems against the background of a decrease in the number of herbivores [53]. Our pedological research in the areas of dune-relief development on the Ob–Tomsk interfluve, no later sand movement has been found to date [36,54,55]. Many researchers, around the world, are finding that aeolian deposits are not necessarily associated with periods of increased wind speed or increased aridity. They can occur during periods of increased sand supply, which is associated with the hydrological cycles of water bodies [56–58]. In the first approximation, we can agree with these conclusions. Indeed, the period of formation of aeolian deposits on the Ob-Tomsk interfluve is nearly synchronized with sand supply along the river valleys flowing from the mountains of the South of Siberia.

### 4.2. Relationships with Modern Soil Formation

Paleosoil 2 differs from the first by a higher content of clay fraction, Feo, and organic carbon, as well as by the abundance of illuviation signs. It is important to note that the signs of illuviation are much less pronounced in modern soils of the aeolian relief within the Ob–Tom interfluve. As a rule, they occur in the peripheral parts of dune fields or when sands and sandy loams are underlain with loam [34,36]. The low content of organic carbon is because the humus horizons of both sections were destroyed by subsequent aeolian processes, as well as by its mineralization.

Soils with lamellae of typical structure that, however, form a single layer with Paleosoil 2 are shown in Figure 9. The lamella that underlies Paleosoil 1, on the slope of the dune, lies inconsistent with the slope. Apparently, aeolian processes destroyed its part during the formation of the blowout. The data on lamellae and their connections with paleosols suggest that lamellae in the sandy soils of the south of Western Siberia are of sedimentary origin and are not a product of illuviation. However, subtler thin lamellae may have an illuvial origin. This is proved by the fact that they restore their continuity according to the breakthrough places with their root canals (Figure 9).

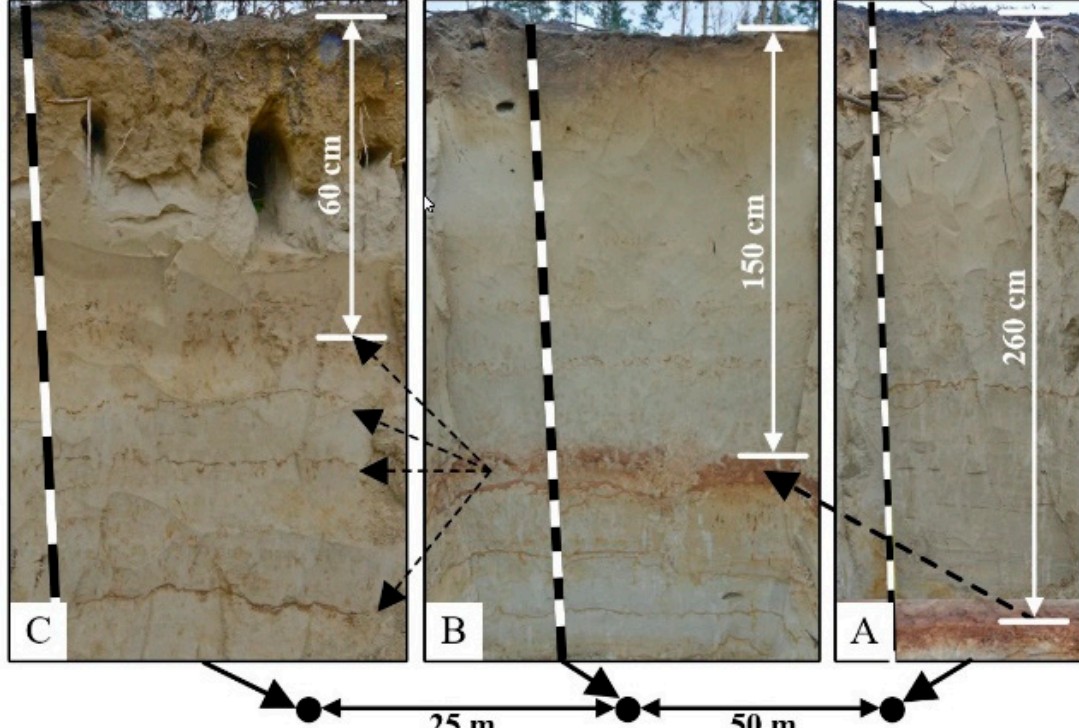

**Figure 9.** Transformation of Paleosol 2 into a series of lamellaes along the slope: (**A**) studied section with paleosol located at a depth of 260 cm; (**B**) appearance of the same paleosol at a depth of 150 cm in the elevated position along the slope; (**C**) series of lamellae at a depth of 60 cm in the most elevated position.

Another interesting consequence is that the soil cover on the sands of the southeast of Western Siberia has a Holocene age. The lithological structure and history of the formation is similar to the territory of the Eastern European Plain. Therefore, the absence of podzols within the dunes' fields of the territory under consideration is due to the peculiarities of continental-soil formation on sands, which prevents podsolization and is not related to the time factor.

### 4.3. Correlation with Other Regions

Two different types of paleosols, typical of periglacial conditions, were found. Both paleosols have a clear stratigraphic position and occur on the border of the aeolian and fluvio-aeolian units similar to Europe and the European territory of Russia. Based on the numerical age of the charcoals from studied soils (Allerød–Younger Dryas), it can be assumed that the formation of inland dunes within the territory of the south of Western Siberia occurred synchronously with similar processes in other regions of Northern Eurasia [3,41,59–62].

The first studied soil can be characterized as an analogue of the Central and Eastern European Usselo horizon corresponding to the Albic Arenosol. Paleosol 1 is distinguished from typical Usselo soil [19] by the absence of a well-defined dark humus horizon rich in charcoals. At the same time, the Eb horizon has a rather large capacity. The first feature can be explained by bioclimatic differences between the regions under consideration, which are expressed today and probably existed in the Late Pleistocene. The modern soils of the areas with aeolian relief of central Poland (Arenosols and Podzols) are characterized by higher organic-matter content and a thicker humus profile [44,63] than Siberian ones, and the average annual temperature is 7 °C higher with similar precipitation. The low-power Ahb of Paleosoil 1 could probably be destroyed because of wind erosion during the activation stage of the aeolian processes.

A.N. Drenova and A.A. Velichko [64] described a soil similar to Paleosol 1 for the European territory of Russia. The research results performed by these authors showed that a pronounced contact of the aeolian pack with the underlying sediments in the form of a thin interlayer of whitish sand (possibly remnants of E horizon) was recorded practically in all studied sections. The situation is similar to the variant described in our research. However, this clarification may be associated with contact gleying. In our case, the arguments in favor of the pedogenic origin of the clarified stratum at the base of the aeolian unit are the presence of separate, low-contrast, Ahb patches, as well as the presence of charcoals in Ahb/Eb, and a sufficiently large depth of the paleosol. In addition, the absence of stratification, the presence of biogenic spotting also speak in favor of the formation of the layer due to soil forming processes. It is important to note that sandy soils of Ob–Tom interfluve layers above lamellae in the C-horizon usually do not have such properties. For example, in the sand overlaying palesoil 2 there is also no contact gleying.

Paleosoil 2 can be identified as an analogue of Finow Soil. The possibility to detect both Finow Soil and Usselo in close locations indirectly confirms that these soils are different elements of a Late Pleistocene paleocatena [19,63].

## 5. Conclusions

The types of paleosoil found in the outcrops of the Takhtamyshevsky quarry can be characterized as analogues of the European paleosoil marker horizons—Usselo soil and Finow soil. The first paleosol, confined to the central part of a small dune, can be identified as Albic Arenosol, the second as Brunic Arenosol (Dystric). Both soils are confined to the boundary of the aeolian unit and underlying sediments and serve as stratigraphic markers. The two radiocarbon ages from the paleosols evidence that the first soil corresponds to the Younger Dryas, and the second is attributed to the Allerød. The results obtained indicate that the activation of aeolian processes in the periglacial zone of the south of Western Siberia took place approximately at the same time as similar processes in other parts of northern Eurasia.

The obtained data may be considered as preliminary results that make it possible suggest that processes that occurred within the southeastern part of Western Siberia were similar to other regions in the periglacial zone of Northern Hemisphere. At the same time, it is not enough for precision paleoenvironmental reconstructions as well as for development of regional chronostratigraphy. Further studies including high resolution OSL dating of reference sections, as well as new observations of paleosols within other areas of Western Siberia where dune fields and aeolian coversends are abundant (Transuralia, Kulunda plain, Tim-Ket interfluve), should help to fill this gap.

**Author Contributions:** A.K. and S.L. performed the fieldwork and soil sampling, conceived and designed the experiments and wrote the paper; A.K., E.K. and G.I. performed chemical analyses; A.N. performed micromorphological investigations; S.K. analyzed the data. All together we discussed the obtained data and corrected the text.

**Funding:** This research was funded by Russian Foundation for Basic Research, grant number 18-34-20129 mol_a_ved.

**Acknowledgments:** We thank Andreas Kirkinis for correcting the English language. We would like to thank the two anonymous reviewers for their constructive comments that improved the manuscript.

**Conflicts of Interest:** The authors declare no conflict of interest. The funders had no role in the design of the study; in the collection, analyses, or interpretation of data; in the writing of the manuscript, or in the decision to publish the results.

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
