# Peer review of "First Findings of Buried Late-Glacial Paleosols within the Dune Fields of the Tomsk Priobye Region (SE Western Siberia, Russia)"

_geosciences, doi:10.3390/geosciences9020082_

Reviewer 1 Report

This is an important paper, so it's important to make sure the ideas within are clearly communicated to an English-speaking audience. I think the issues I have with this paper are primarily issues with translation to English with the additional issue that soils are described with different terminology in the US (where I do my research) and in Russia. 

I have uploaded a file (review_geosciences-422137.docx) with my specific comments.

Author Response

The manuscript received very constructive reviews that allowed greatly improving our interpretation and presentation of results. The authors thank the reviewers and the editors of the journal for their work on the text of the article and for giving us an opportunity to revise this paper.

The comments received from the reviewers certainly helped to improve the text of the article. Detailed response to reviewers’ comments is given below. We agree with all of them and in most cases we incorporated their suggestions in the revision. Most of the specific comments on the text were corrected in accordance with the proposals. All the changes in revised manuscript are highlighted.

While revising the paper we have tried to move away from concrete conclusions towards more flexible assumptions and hypotheses (in accordance with the proposals of Reviewer 2). We agree that more tentative interpretation and more careful conclusions are necessary, since this paper presents preliminary studies, the number of dates is indeed small, and the Quaternary geology of the Ob-Tom region is relatively poorly studied. The aim of the work was the characterization of the first findings of paleosols within the region and their comprehension with the analogies from Central and Eastern Europe, so we did not set a goal to make far-reaching conclusions based on the available data. Issues related to the geomorphology of the dunes, their distribution, and the vegetation that existed during the period under review are mainly based on a limited published data devoted to the study of bottom sediments and peatlands within the region (questions from Reviewer 1) and probable should be a part of a more comprehensive regional geomorphological and paleoenvironmental studies. We also tried to make conclusions about timing of soil development more flexible and agree that only high-resolution OSL-dating complement by detailed sedimentological studies (heavy and light mineral analysis and quartz grains morphoscopy) would make it possible to create highly reliable regional chronostratigraphy. However, such research is more likely related to geomorphology and quaternary geology than paleopedology.

Some terms (for example, morphons) were replaced. Russian-language sources in the bibliography were preserved, since they are important for understanding the soil, geomorphological, and paleoenvironmental context of the territory and, therefore, deserve mention (Reviewer 2 suggestions). The author consider that it possible not to give superfluous information related to the peculiarities of the European paleo-horizons, since they are described in details in the review paper by Kaiser et.al. (2009) (Proposal of Reviewer 1). The authors are familiar with the literature devoted to aeolian sand deposits and Late Pleistocene aeolian processes for the territory of North America. At the same time, in this preliminary work, we tried to compare the obtained results with more comprehensible for us and close in terms of geography territories of Eastern Europe. We fully agree with the idea proposed by Reviewer 1, so it was added to the discussion of the article.

Several proposals devoted to further areas of research that will allow us to obtain a more detailed description of the conditions that existed in the region during the period under review were added to the conclusions (Proposal of Reviewer 2).

We thank three reviewers for very constructive remarks.

Care of these and other self-motivated corrections, we feel that our revised manuscript can meet the high standards of the journal. Hope to hear from you soon,

Corrections are listed in the attachment.

Reviewer 2 Report

Generell:

This is a very interesting manuscript for a broad audience of Quaternary scientists and paleopedologists. The authors investigated a so far internationally unknown outcrop which may in future be designed an Eurasian key section of Late Glacial to Holocene paleoecological development. Some minor revisions and/or improvements are, however, necessary. They include in particular linguistic corrections and clarifications and a more cautious discussion and interpretation of the chronostratigraphic position of the two investigated paleosols. Even if the suggested chronology appears most likely it basically relies on only two calibrated 14-C ages requiring more chronological data. With the ages available so far the chronostratigraphy can only be tentative. Furthermore its discussion and interpretation should clarify the distinction between the time duration of soil formation and a singular dated event (burning of wood) which most probably occurred within or at the end of this time duration. From this viewpoint the conclusions should emphasise the necessity and aims of further research.

Many (18 from 60) cited references are in Russian and, thus, hardly or not accessible to an international audience. Although some fundamental publications in Russian need to be cited the authors may consider limiting themselves to essential publications in Russian. Also several other cited publications in English may be reconsidered if they are all relevant for understanding the circumstances in the study area.

Details:

For minor errors (type errors, missing articles, etc.) see the annotations in the manuscript PDF.

-          Line 32: You suggest that both soils correspond to the European Usselo Soil? If this is correct, state more clearly because at the first glance lines 31 to 33 may look like a mismatch.

-          Line 44: replace “lake glacial” by glaciolacustrine, and “lake alluvial” just by alluvial.

-          Line 51: the term “rappers” is not suitable in this context; replace by a more appropriate word.

-          Line 100, Figure 1.and line 133: “career” is a wrong word. Do you mean outcrop or quarry?

-          Line 130, 312, 417: “absolute age” is misleading as all measurements have an error. Replace by numerical age.

-          Figure 3: no proper scientific citation of calibrated radiocarbon ages. Give the confidence intervals (from… to…). Ternary diagram axis: less than 0.002 mm. Legend: lamella.

-          Line 231, 242, 343: “heavier granulometric composition” is confusing. Better: “granulometric composition of higher density” or “higher density granulometric composition”.

-          Table 2 headline, H2O: 2 as subscribe character H2O.

-          Line 280: “illumination” does not make sense here; you probably mean illuviation.

-          Line 281: I recommend “places” instead of “areas”.

-          Line 292: “weed” is not an appropriate word. Replace.

-          Line 312-314: This statement is not justified! The age just says that the tree grew in the middle of the younger Dryas. Soil formation is a longer lasting process which cannot unambiguously be derived from just one charcoal age. As physical dating methods (radiocarbon, OSL, …) determine the age of an event the duration of soil formation can only be estimated by age bracketing, i.e. several ages spanning the time of soil formation. This is critical for radiocarbon because organic remnants preferably origin from the end of soil’s exposure to the surface, but they may also be redeposited (although unlikely in this case). In order to more reliably determine the time span of soil formation bracketing OSL ages (dating the last exposure to daylight) from the youngest sediments below and the oldest above a buried soil should complete calibrated radiocarbon ages. One radiocarbon age from each soil anyhow cannot tell this, but must rather be interpreted as a hint to the age of the soil.

-          Line 349-350: Syntax error. Do you mean “…rises practically to the surface”?

-          Line 352: Not an unambiguous interpretation. Designate it as tentative interpretation.

-          Line 356: “Holocene”: why not during the Allerød; see line 396.

-          Line 364: “persisted” would be more appropriate than “existed”.

-          Line 369: replace “prove” by “support”.

-          Linue 373: What means “even-edged”? I could not find out a meaning of this term. Can you explain or replace it?

-          Line 385: What means “solid runoff”? Do you mean runoff with transport of solids? Express clearly.

-          Lines 434: What means “mountains E”? Explain clearly or use a better term.

-          Lines 452-454: Rephrase the full sentence, e.g.: “The two radiocarbon ages from the paleosols evidence that the first soil corresponds to the Younger Dryas, and the second is attributed to the Allerød”.

-          Conclusions: They are too scarce. Conclusions may include the need for further research such as more refined stratigraphy along a broader (and partly deeper) exposure, and, in particular, more age determinations by radiocarbon and Optically Stimulated Luminescence. By this means the site may be designed a Northern Eurasian key section.

Author Response

(The authors gave the same response as above.)

Reviewer 3 Report

The text is appropriated for the journal. The methodology and results support the discussions and conclusions.

Author Response

The manuscript received very constructive reviews that allowed greatly improving our interpretation and presentation of results. The authors thank the reviewers and the editors of the journal for their work on the text of the article and for giving us an opportunity to revise this paper.

The comments received from the reviewers certainly helped to improve the text of the article. Detailed response to reviewers’ comments is given below. We agree with all of them and in most cases we incorporated their suggestions in the revision. Most of the specific comments on the text were corrected in accordance with the proposals. All the changes in revised manuscript are highlighted.

While revising the paper we have tried to move away from concrete conclusions towards more flexible assumptions and hypotheses (in accordance with the proposals of Reviewer 2). We agree that more tentative interpretation and more careful conclusions are necessary, since this paper presents preliminary studies, the number of dates is indeed small, and the Quaternary geology of the Ob-Tom region is relatively poorly studied. The aim of the work was the characterization of the first findings of paleosols within the region and their comprehension with the analogies from Central and Eastern Europe, so we did not set a goal to make far-reaching conclusions based on the available data. Issues related to the geomorphology of the dunes, their distribution, and the vegetation that existed during the period under review are mainly based on a limited published data devoted to the study of bottom sediments and peatlands within the region (questions from Reviewer 1) and probable should be a part of a more comprehensive regional geomorphological and paleoenvironmental studies. We also tried to make conclusions about timing of soil development more flexible and agree that only high-resolution OSL-dating complement by detailed sedimentological studies (heavy and light mineral analysis and quartz grains morphoscopy) would make it possible to create highly reliable regional chronostratigraphy. However, such research is more likely related to geomorphology and quaternary geology than paleopedology.

Some terms (for example, morphons) were replaced. Russian-language sources in the bibliography were preserved, since they are important for understanding the soil, geomorphological, and paleoenvironmental context of the territory and, therefore, deserve mention (Reviewer 2 suggestions). The author consider that it possible not to give superfluous information related to the peculiarities of the European paleo-horizons, since they are described in details in the review paper by Kaiser et.al. (2009) (Proposal of Reviewer 1). The authors are familiar with the literature devoted to aeolian sand deposits and Late Pleistocene aeolian processes for the territory of North America. At the same time, in this preliminary work, we tried to compare the obtained results with more comprehensible for us and close in terms of geography territories of Eastern Europe. We fully agree with the idea proposed by Reviewer 1, so it was added to the discussion of the article.

Several proposals devoted to further areas of research that will allow us to obtain a more detailed description of the conditions that existed in the region during the period under review were added to the conclusions (Proposal of Reviewer 2).

We thank three reviewers for very constructive remarks.

Care of these and other self-motivated corrections, we feel that our revised manuscript can meet the high standards of the journal.

Round  2

Reviewer 1 Report

I am satisfied with the revisions.